# ACE: A Cross-Platform Visual-Exoskeletons System for Low-Cost Dexterous Teleoperation

**Shiqi Yang**    **Minghuan Liu**    **Yuzhe Qin**    **Runyu Ding**    **Jialong Li**
**Xuxin Cheng**    **Ruihan Yang**    **Sha Yi**    **Xiaolong Wang**

UC San Diego

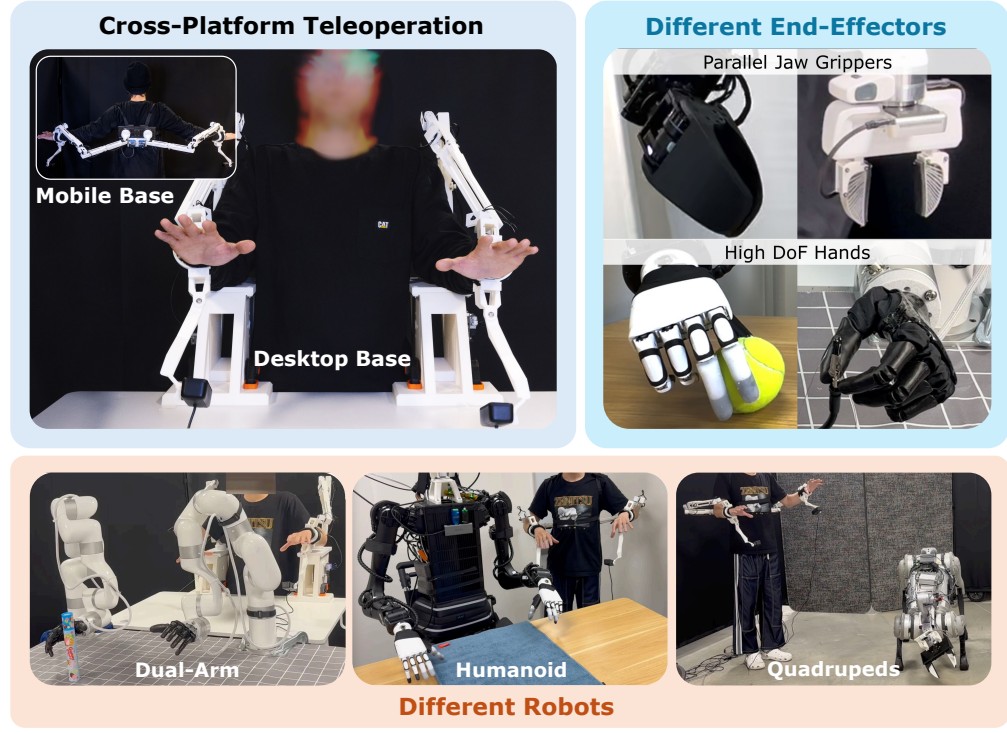

Figure 1: **An Overview of the Proposed** `ACE` **System.** The system consists of two bimanual exoskeleton arms and two cameras for hand pose tracking. Together with our modular design of the base, we can perform teleoperation across a wide range of end effectors and robot platforms.

**Abstract:** Learning from demonstrations has shown to be an effective approach to robotic manipulation, especially with the recently collected large-scale robot data with teleoperation systems. Building an efficient teleoperation system across diverse robot platforms has become more crucial than ever. However, there is a notable lack of cost-effective and user-friendly teleoperation systems for different end-effectors, e.g., anthropomorphic robot hands and grippers, that can operate across multiple platforms. To address this issue, we develop `ACE`, a cross-platform visual-exoskeleton system for low-cost dexterous teleoperation. Our system utilizes a hand-facing camera to capture 3D hand poses and an exoskeleton mounted on a portable base, enabling accurate real-time capture of both finger and wrist poses. Compared to previous systems, which often require hardware customization according to different robots, our single system can generalize to humanoid hands, arm-hands, arm-gripper, and quadruped-gripper systems with high-precision teleoperation. This enables imitation learning for complex manipulation tasks on diverse platforms. Webpage: https://ace-teleop.github.io/.

8th Conference on Robot Learning (CoRL 2024), Munich, Germany.

# 1 Introduction

In recent years, the effectiveness of training robot foundation models using real-world robot data has led to a significant increase in attention to data collection. Teleoperation has emerged as a crucial data collection method, enabling researchers to demonstrate and record complex robotic tasks. Various solutions, including VR systems [1, 2, 3], motion capture systems [4, 5], wearable gloves [6, 7], and low-cost devices [8, 9], have been developed to facilitate teleoperation, each offering unique advantages in terms of accessibility, precision, and generalizability. As we assess these existing systems, a critical question arises: *What key information must be captured to effectively perform dexterous manipulation tasks on a wide range of robot platforms*?

The answer is quite simple: accurate hand poses and end-effector positions, which can be achieved with low-cost hardware. We propose ACE, a cross-platform visual-exoskeletons system for low-cost dexterous teleoperation, as illustrated in Figure 1. For hand pose capture, we utilize a low-cost camera combined with a 3D hand pose estimation method [10, 11, 12]. To address the common issue of occlusion on vision-based systems, our design features a camera mounted on the end effector of the exoskeleton, ensuring it always follows the front face of the hand. The exoskeletons, which are 3D printed, provide precise end-effector positions through forward kinematics. With this, the robot hardware does not need to match the morphology of the teleoperation system. During teleoperation, as the operator moves the exoskeletons and their hands, we capture the hand root end-effector position and hand pose in real-time. We then use inverse kinematics to map the robot arm end-effector with the human hand end-effector position. This allows for effective motion retargeting between human hands and robot hands, enabling precise control of the robot's fingers.

Table 1: Comparison of Teleoperation Systems (EE: End Effector, FK: Forward Kinematics)

| Teleop System | EE Type | $ Cost | EE Tracking | Cross Platform? | Mobile Base? |
|---|---|---|---|---|---|
| ALOHA [8] | Parallel-Jaw | 20k | Joint-matching | ✗ | ✗ |
| Mobile-ALOHA [13] | Parallel-Jaw | 32k | Joint-matching | ✗ | ✓ |
| GELLO [9] | Parallel-Jaw | 0.6k | Joint-matching | ✗ | ✗ |
| AnyTeleop [14] | Anthropomorphic | ∼ 0.3k | Vision | ✓ | ✗ |
| DexCap [15] | Anthropomorphic | 4k | Hand Mocap + Vision | ✓ | ✓ |
| ACE (Ours) | Both | 0.6k | Vision + FK | ✓ | ✓ |

We compare our teleoperation system with several recently proposed systems, as shown in Table 1, to illustrate our advancement. The **ALOHA** [8, 13] system provides precise control of bimanual manipulators, primarily focusing on parallel-jaw grippers. By directly matching the joints between the teleoperation system and the robot, ALOHA can achieve precise control of the end-effector. However, this advantage limits its usage to only the specific designated hardware. In contrast, our system allows the operation of multiple robot hardware types and offers the flexibility to control multi-finger robot hands instead of only parallel-jaw grippers. This flexibility enables data collection on more diverse tasks. Our system can even be adapted to 2-DOF grippers using simplified hand gestures. The **GELLO** [9] system achieves one-to-one joint matching by using a smaller-sized robot arm to control the actual robot arm. However, we show in our experiments that this mismatch in size makes it hard to control for fine manipulation tasks. Our system, on the contrary, provides more precise control by focusing on transferring the motion of only the end-effectors. **AnyTeleop** [14] uses cameras to capture hand poses and perform teleoperation directly with the camera data. While this method pushes the boundary of low cost and simplicity in teleoperation hardware, the vision-based system can only provide limited accuracy in measuring the hand root end-effector poses. Our system, which employs an exoskeleton for capturing hand poses, provides a more accurate and reliable measurement of the hand root end-effector.

Our system leverages the precision of kinematics-based exoskeletons, combined with the affordability and adaptability of vision-based systems. Acting as a bridge between these two approaches, we can obtain accurate hand poses and end-effector positions at a low cost while ensuring cross-platform compatibility. In this paper, we introduce our teleoperation system, including the hardware design and the software modules that enable cross-platform operations. Our experiments show two key advantages. First, our designed system enables users to quickly adapt and efficiently, accurately,

and swiftly complete tasks under various precision and workspace requirements. Second, our imitation learning experiments demonstrated that our system can efficiently collect effective data and complete the corresponding imitation learning.

## 2    Related Work

**Learning from human demonstrations.** Learning from human demonstrations has emerged as a widely adopted approach in robotic manipulation, leveraging the ability to replicate complex tasks by observing human actions. In-the-wild data from human hand movements offers a rich and accessible source of demonstration data for robot learning [2, 16, 17, 18, 19, 20]. Recent studies have shown that utilizing teleoperation for robot control and data collection is both efficient and transferable [13, 14, 9, 21, 22]. However, collecting large-scale datasets in real-world environments is often time-consuming and resource-intensive [23, 24, 25, 26]. To enhance efficiency and expedite data collection, it is crucial to develop systems that are generalizable to a wide range of robot platforms, end-effectors, and tasks.

**Vision-based Teleoperation.** Teleoperation for manipulation has a long history [27], with common solutions like VR [2, 1, 28, 29, 30, 31], motion capture devices [4, 5], and gloves [6, 7], smartphones [32, 33], and other devices [34, 35]. Those systems provide accurate tracking poses for manipulation tasks. However, high-cost devices are less accessible, which limits large-scale data collection. In contrast, vision-based teleoperation is particularly adaptable and user-friendly [36, 37, 38], which requires less hardware setup, can be adapted to different body types, and is relatively low-cost without complex hardware setups. Effective hand-tracking algorithms [10, 11, 12] allow for the transfer of human hand motions to anthropomorphic robot hands [39, 14, 40, 38]. Direct wrist tracking and motion transfer allow for cross-platform operations on a wider range of robot platforms [3, 18, 39, 41, 42]. However, vision-based tracking systems are sensitive to occlusion, which introduces ambiguities in the hand poses. Wrist pose estimation in the world coordinate can also be inaccurate and noisy.

**Kinematics-based Data Collection.** Data collection directly from the robot itself has also been employed [43]. Recent works of developing a replica of a robot have shown high efficiency for data collection [44] and teleoperation [8, 9]. These systems mirror the target robot's kinematic structure, enabling accurate movement and interaction replication. The controlling process is straightforward with joint-matching [13, 9, 45]. Exoskeletons are also intuitive solutions for human operators, but normally with a significantly higher cost [46, 47, 48, 49] and do not support hand modeling. Our work is also related to recently proposed DexCap [15] (Table 1) which utilizes a hand motion capture system with a mobile base, with more expensive hardware. Its wrist tracking is based on real-time localization, which is more accurate than pure vision methods but introduces delay. Compared to DexCap, our system allows cross-platform generalization with enhanced accuracy and flexibility using a low-cost design.

## 3    System Design

To develop a cross-platform and low-cost dexterous teleoperation system that enables efficient large-scale data collection, we summarize the desiderata into the following five principles: **Cross-platform Compatibility**: Ensuring versatility across various robot platforms, such as fixed-based robot arms, quadrupeds, and humanoids, while accommodating different end-effector types, including anthropomorphic hands and parallel-jaw grippers. **Accuracy**: Transferring the precise hand and wrist pose of the operator to actual robot platforms. This enables fine-grained manipulation tasks and improves the success rate of data collection. **Low-cost**: Emphasizing affordability to facilitate easy prototyping and lower the barrier to large-scale data collection. This encourages the use of readily available, off-the-shelf components and materials, instead of expensive components like VR headsets or motion tracking systems. **User-friendly**: Ensuring the system is intuitive to use, requiring minimal expertise for calibration and operation, and accommodating users of different heights and weights. We also aim for two types of bases - fixed or mobile - so the user can easily switch between fixed and mobile tasks. **Easy to Manufacture and Maintain**: Designing the

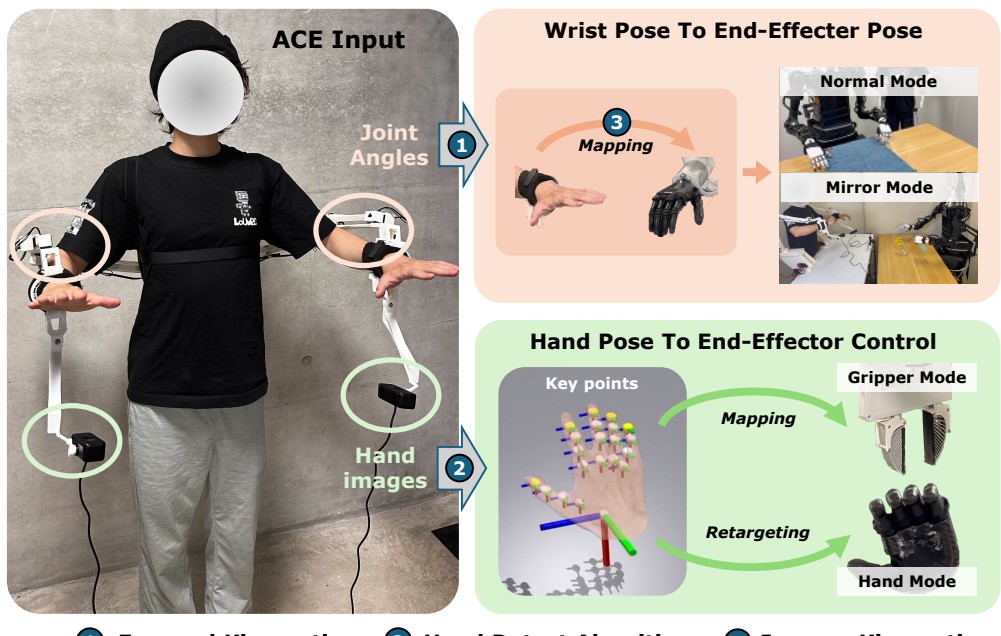

Figure 2: **Architecture of the ACE Teleoperation System.** Our system reads the joint angles from our exoskeleton motors and the hand image to estimate the wrist and hand poses through forward kinematics and a hand detection algorithm. With different modes of operation, we can perform teleoperation on different end-effectors and robot platforms.

system to be straightforward to manufacture and maintain, with easily replaceable components to minimize downtime and extend operational lifespan. This includes a modular design using standard components to support easy assembly, disassembly, and replacement.

An overview of our ACE system design is illustrated in Fig. 2. Our system consists of a 3D-printed bimanual exoskeleton and two hand-facing low-cost webcams mounted at the ends of the exoskeleton. The exoskeleton's joint position tracking provides accurate wrist pose by forward kinematics, and the two cameras provide precise hand-pose tracking. By combining these two sources of information, the system enables real-time hand pose tracking relative to the world coordinate.

After obtaining the accurate hand pose, we map it to target robot poses on various robot platforms. After the control mapping, we can teleoperate robots and collect demonstrations for imitation learning. In experiments, we demonstrate the effectiveness of our teleoperation system on a range of robots with various end-effectors, including the Xarm with Ability Hand, H1 with Inspire Hand, B1 with Z1, Franka with a gripper, and GR-1 with a gripper.

## 3.1 Hardware Design

**Dual-base setup.** The proposed ACE system, shown in Fig. 3, has two arms and two types of bases: desktop and mobile. This dual-base setup provides cross-platform compatibility from mobile robots to robot arms. The desktop version provides a stable and precise wrist pose from its stationary base. For long-term teleoperation, the user may rest their elbows on the platform. The mobile base is designed for tasks that need constant movement or continuous adjustment of the field of view. Specifically suitable for mobile platforms including humanoids and quadrupeds.

**Servos and cameras.** Our system features two arms, each with seven links, six degrees of freedom (DoF), a wrist, and a camera mount. Each arm is equipped with a UCB2Dynamixel (U2D2) controller and DYNAMIXEL XL330-M288-T servos with high-resolution 12-bit encoders for accurate joint position readings, ensuring precise end-effector tracking. The two-DoF camera mount allows for optimal hand positioning to avoid occlusion issues common in vision-based hand tracking.

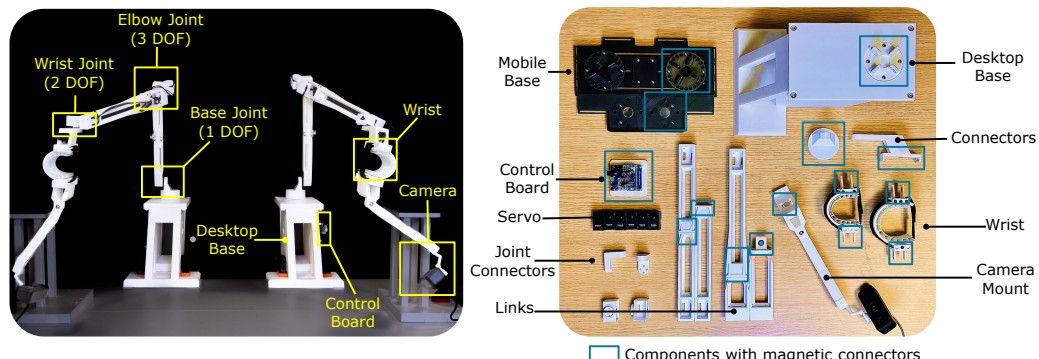

Figure 3: **Hardware Components.** Left: assembled exoskeleton on a fixed desktop base. Right: parts for one arm. We show two wrist connectors and links of different sizes.

**Magnetic connections.** We utilize a modular design with magnetic connections, as illustrated in Fig. 3. The use of magnets is not only inexpensive but also significantly enhances the user experience. Traditionally, donning exoskeleton equipment can be a complex process. However, our magnetically modular design allows users to independently don the system (the desktop version) in less than 30 seconds. Additionally, the magnetic connections facilitate quick and easy size adjustments. The wrist size can be altered almost instantly, and the upper and lower arm length adjustments can be completed in under 2 minutes. This modular design allows easy integration of updated components, enabling efficient upgrades to our open-source system.

### 3.2 Pose Estimation and End-Effector Control Mapping

When the user starts collecting demonstrations, we estimate the precise wrist pose and hand keypoints. The wrist pose is determined by reading joint angles from the exoskeleton's encoders and calculating the pose using the forward kinematics of the designed exoskeletons [50]. Hand images are processed using MediaPipe [11], a lightweight, RGB-based hand detection tool that can operate in real-time on a CPU, to detect the 21 hand keypoints in the wrist frame.

After obtaining the wrist and hand pose data, we map human movements to target robot poses and end-effector control signals across various platforms. To meet the specific needs of different platforms and tasks, we address three significant challenges in developing the control interface: workspace mismatch, control scale variability, and usability concerns.

**Matching workspace sizes.** First, the variation in workspace sizes poses a significant challenge on different platforms, particularly in achieving full coverage during bimanual operations. For a given platform, users can specify the required workspace in the configuration. By wearing our device and moving their arms around, we align the center of the human and robot workspaces and document the matching scale between these two workspaces.

**Control scale variability.** Second, different tasks and platforms require varying levels of control precision. For example, when using a scaled replica of the actual robot platform, direct joint-matching teleoperations will result in errors being amplified when transferring the motion to a larger robot arm [9]. Moving the small teleoperation end-effector by a small distance may translate to much larger movement on the actual robot arm, significantly impacting control accuracy for fine tasks. Instead, in our system, by using inverse kinematics to map the end-effector positions, we can provide a more accurate and intuitive mapping from the teleoperation device to the actual robot. Consider the 6D robot end-effector pose $\mathbf{x}_e \in \mathbb{R}^6$, and the human wrist pose $\mathbf{x}_h \in \mathbb{R}^6$, we map $\mathbf{x}_h$ to $\mathbf{x}_e$ as

$$\mathbf{x}_e = \gamma(\mathbf{x}_h - \mathbf{c}_h) + \mathbf{c}_t \tag{1}$$

where $\gamma$ is the control scale, $\mathbf{c}_h$ is the center of the human workspace, and $\mathbf{c}_t$ is the center of the task workspace. More detailed parameter definitions are included in the Appendix A. We show that in our experiments, this scaled approach is more efficient and adaptable to different workspaces and task sizes than direct joint matching methods.

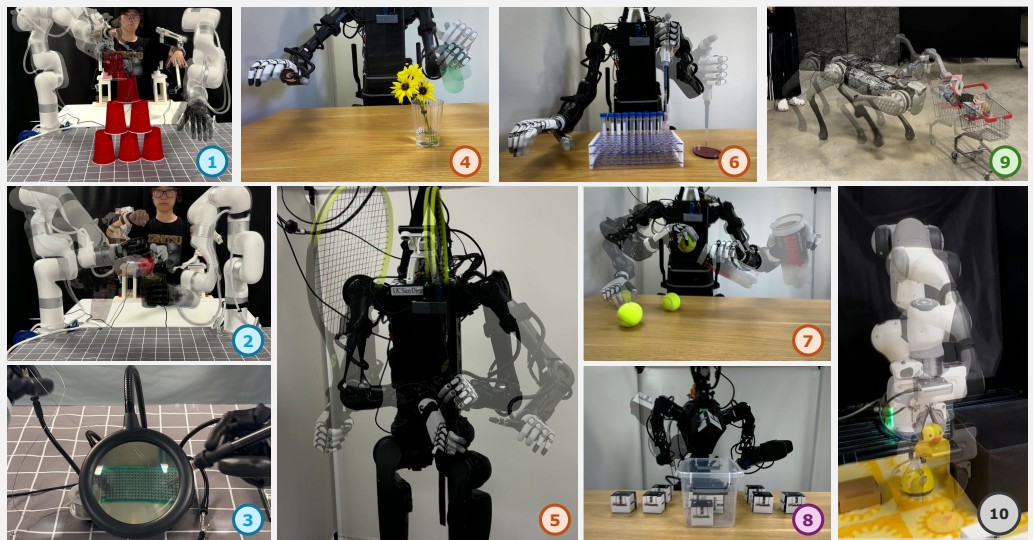

Figure 4: **Examples of Cross-Platform Teleoperation.** Examples 1-3 are performed on the *xArm with ability hand* setup: 1) stacking, 2) serving coffee, and 3) soldering. Examples 4-7 are executed on the *H1 with inspire hand* setup: 4) spraying, 5) passing, 6) pipetting, and 7) inserting tennis. Example 8 is on the *GR-1 with a gripper*, performing a box-packing task. Example 9 features the *B1 with Z1* setup, demonstrating a shopping cart-pushing task. Example 10 is on the *Franka with a gripper*, performing a task of picking up miscellaneous objects.

**Usability.** The biggest challenge for cross-platform teleoperation is usability. Cross-platform deployment often involves redundant and labor-intensive calibration processes when switching platforms. To enhance usability and accommodate different tasks and robot platforms, we have different control modes as an additional layer before transferring the end-effector pose to the actual robot:

- *Normal Mode*: Direct transfer end-effector positions to the actual robot. The end-effector pose in this mode $\mathbf{x}_e^{normal}$ is the same as in Eq. (1), where $\gamma$ is the ratio between the robot and human workspace radius.

- *Mirror Mode*: Designed for large robots, this mode allows users to operate face-to-face with the robot. In this mode $\mathbf{x}_e^{mirror} = -\gamma_e^{mirror}(\mathbf{x}_h - \mathbf{c}_h) + \mathbf{c}_t$ where the end-effector motion is mirrored, thus making it more intuitive during operation.

- *Bimanual Mode*: For bimanual tasks that require high precision, the cameras may collide. We adjust the scaling factor $\gamma$ and $c_t$ so that the left and right hand side workspaces align. In this setting, when the camera mounts are about to collide, the robot hands touch each other.

Based on different types of end-effectors, we also support:

- *Gripper Mode*: Tailored for parallel jaw grippers, this mode provides straightforward control for tasks suited to this end-effector type. The distance between the thumb tip and the index fingertip is linearly mapped to a range of 0 to 1.

- *Hand Mode:* The hand and fingertip motions are retargeted to the robot hand; detailed principles are provided in the Appendix B.

After mapping the wrist poses to the robot's coordinate frame, the joint angles of the robot arm can be obtained using inverse kinematics (IK). Different robot platforms may require specific filters, PID parameters, and constraints to ensure safety, smooth movements, and reduce issues from singularity. The robot hand joints are obtained from human hand poses and mapped through hand motion retargeting [14]. We have optimized the computational efficiency to ensure a hand pose tracking frequency of approximately 27 Hz, which can be further improved to 100 Hz with a higher-frequency camera. These approaches ensure that our teleoperation system is adaptable, precise, and

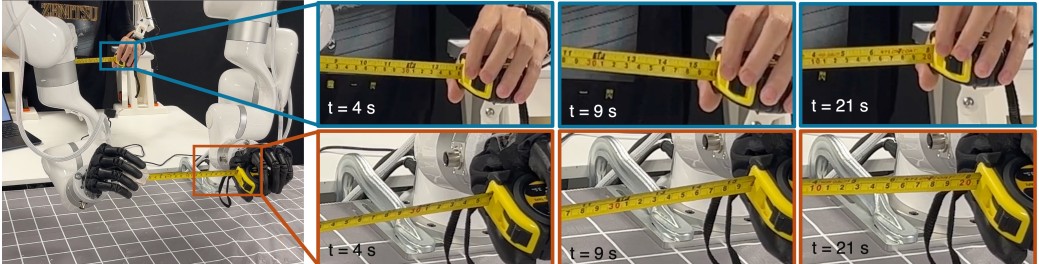

Figure 5: **End Effector Accuracy Evaluation.** By extending the tape from 20 cm to 40 cm, the teleoperation end-effector demonstrates precise replication of the operator's motion with an average error of only 3 mm.

user-friendly, addressing the primary challenges of deploying such systems across diverse robotic platforms.

## 4   Experiments

We designed experiments to investigate two key aspects of our proposed teleoperation system: precision and transferability. We designed a target-reaching task to show that our system can achieve precise end-effector control, compared to previous joint-matching systems. We show how robots autonomously perform tasks via imitation learning trained using the data collected from our system, and we show more teleoperation examples in Fig. 4.

### 4.1   End Effector Accuracy and Efficiency

**Precision evaluation.** Similar to other joint matching methods [51, 45, 9], the end-effector pose obtained from the joint encoder is highly accurate. Specifically, in our system, the average forward kinematics error is around 1 mm. As shown in Fig. 5, we perform teleoperation on a measurement tape extension task. By extending the tape from 20 cm to 40 cm, we show that the teleoperation end-effector accurately represents the operator's motion, with an average error of only 3mm. The errors presented in this figure are cumulative, including both teleoperation and robot control errors, yet the overall error remains within a reasonable range.

**Target-reaching evaluation.** We compare the agility and precision of our ACE system with the previous joint-matching system, specifically *GELLO* [9] for its comparable cost and open-sourced implementation. We designed a target-reaching task to evaluate the performance of both systems in a simulation. The operator's objective was to control the end-effectors to reach randomly generated target regions. Success was recorded when both end-effectors reached their respective target area and remained within the target regions for a short duration.

We set up four scenarios to assess teleoperation performance, with workspaces and target regions of small, medium, and large sizes. Our evaluation involved six operators. Further details about the task configurations and operator specifics are provided in Appendix C.3. We evaluate the performance of the system by the time used to reach the target region, the average reaching speed, the average end-effector moving speed, and the effective ratio, which is the ratio between the optimal moving distance and the actual moving distance of the end-effector.

The result is shown in Table 2. ACE system consistently outperformed GELLO across most scenarios and metrics, demonstrating shorter time costs, higher reaching speeds, higher effective ratios, and success rates. Because of the combination of using Forward Kinematics (FK) for end-effector tracking, and Inverse Kinematics (IK), allows for adaptable mapping from the operator pose to the robot pose. Although the direct joint-matching method in GELLO provides accurate control, it is only suitable for a specific workspace and task size (medium size as defined in our experiment). For smaller workspaces, or fine tasks that require precise motion, the amplified robot motion in GELLO has a hard time stabilizing to the given position, resulting in redundant or sub-optimal motion. Our

Table 2: **User Study Evaluation in Simulation.** Comparison of ACE with GELLO baseline in a target-reaching task across varying workspace and target sizes. The results demonstrate ACE's superior performance in terms of reach time, velocity, effective ratio, and success rate, particularly in small and medium workspace scenarios. Even in a large workspace, ACE consistently outperforms GELLO, showing its robustness across different conditions.

| Workspace | Target | Method | Avg Reach Time (s) ↓ | Avg Reach Vel (m/s) ↑ | Avg EE Vel (m/s) ↑ | Effective Ratio ↑ | Success Rate ↑ |
|---|---|---|---|---|---|---|---|
| Small | Small | GELLO | 13.6 | 0.089 | 0.340 | 26.2% | 47.6% |
| | | ACE | **4.69** | **0.220** | **0.360** | **61.1%** | **97.1%** |
| Medium | Small | GELLO | 6.52 | 0.179 | 0.395 | 45.3% | 73.8% |
| | | ACE | **4.54** | **0.249** | **0.401** | **62.1%** | **91.4%** |
| Medium | Medium | GELLO | **3.65** | **0.327** | 0.436 | **75%** | **93.8%** |
| | | ACE | 4.05 | 0.294 | **0.438** | 67.1% | 87.5% |
| Large | Large | GELLO | 6.52 | 0.212 | 0.491 | 43.2% | 68.7% |
| | | ACE | **5.17** | **0.264** | **0.504** | **52.6%** | **78.6%** |

system ACE utilizes an adaptable approach that first maps the operator's natural workspace to the task's required workspace, and also adjusts the control scale on the robot platform to fit the required tasks and workspaces. This adaptable approach is only available when we can transfer the scaled poses with inverse kinematics to solve the robot configurations. This shows that our approach is not only more generalizable to different robot platforms and end effectors but also adaptable to tasks and workspaces of different scales.

## 4.2 Imitation Learning

We design six real-world tasks that require multi-step dexterous teleoperation, illustrated in Fig. 6. Two long horizon tasks, *Vacuum Keyboard*: 1) grasp the vacuum, 2) activate the vacuum, 3) clean the keyboard, 4) return the vacuum; *Serve Candies*: 1) grasp the candy tube, 2) open the cap, 3) throw the cap into the trash bin, 4) pour candies into the box, and 5) throw the tube into the trash bin. *Wipe Whiteboard*: grasps the eraser and wipes the whiteboard. *Grasp Dolls*: grasps the dolls one-by-one and put into the basket. *Put in Tennis*: grasp a tennis ball and put it into the container. For *Grasp Dolls*, it requires generalizations to different objects and robot positions. In particular, we trained on nine types of small dolls and two types of large dolls but tested on 15 types of small dolls and three types of large dolls. We collected data with ACE system on xArm with Ability Hand and H1 with Inspire Hand on the above-mentioned tasks. Tab. 3 shows that with ACE, we can efficiently collect a large number of data with a high success rate.

With the data collected from teleoperation, we train each policy with imitation learning. We use 3D diffusion policies [52] for imitation learning on xArm platforms and ACT [8] on H1. Specifically, for diffusion policy, we draw point clouds independently from two camera views, and the proprioception states as the robot observation and the action space is the target poses of the arm wrists, and hands. The input for ACT is a single RGB image from a camera mounted at the top of the robot. The learning results for all tasks are shown in Tab. 3. We tested 10 episodes for each task and showed the success rate on different tasks.

## 5 Conclusion

In this paper, we proposed ACE: a cross-platform visual-exoskeleton system for low-cost dexterous teleoperation. ACE system employs a hand-facing camera to capture 3D hand poses and an exoskeleton mounted on either a fixed or a mobile base. This setup accurately captures both the hand root end-effector and hand pose in real-time, enabling cross-platform operations with low-cost hardware. Our system does present two major limitations. The camera mount prevents the wearer's hands from coming together and, additionally, increases the burden of rotational movements due to the added moment arm. Although functionally addressed by our scaled controller, certain operations are less intuitive.

**Acknowledgments**

This work was supported, in part, by the Qualcomm Innovation Fellowship and the Technology Innovation Program (20018112, Development of autonomous manipulation and gripping technology using imitation learning based on visual and tactile sensing) funded by the Ministry of Trade, Industry & Energy (MOTIE), Korea. We thank Xuanbin Peng, Chengjing Yuan, Mazeyu Ji, Chenhao Lu, Haichuan Che, Jiyue Zhu, and Rizhao Qiu for their generous help on experiments and related videos.

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

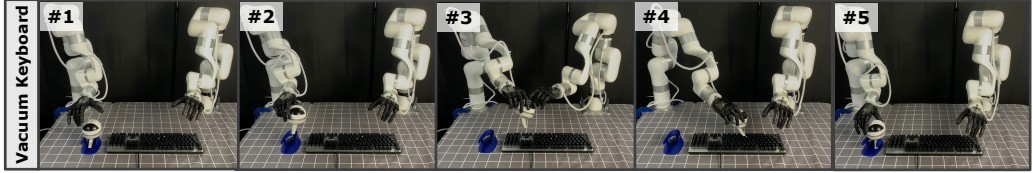

(a) The robot grasps the vacuum (*#1, #2* - Grasp), activates it by pressing a button (*#3* - Activate), uses the vacuum to clean the keyboard (*#4* - Clean), and then returns the vacuum to its original position (*#5* - Place).

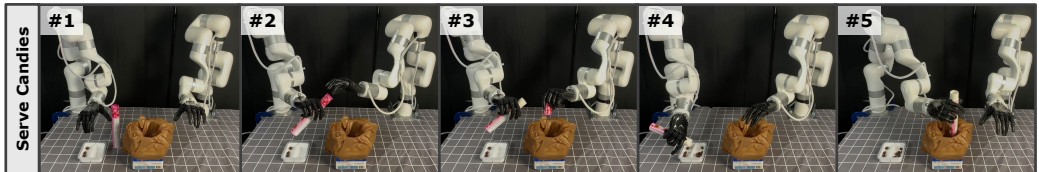

(b) The robot grasps the candy jar (*#1* - Grasp), opens the jar by removing the lid (*#2* - Open), discards the lid (*#3* - Throw Cap), pours out the candies (*#4* - Pour), and finally discards the jar (*#5* - Throw Tube).

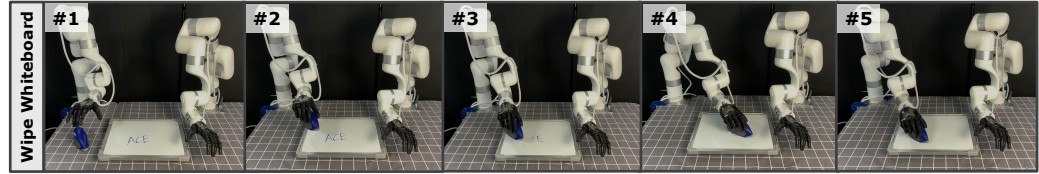

(c) The robot grasps the eraser (*#1* - Grasp) and then wipes the whiteboard (*#2-#5* - Wipe).

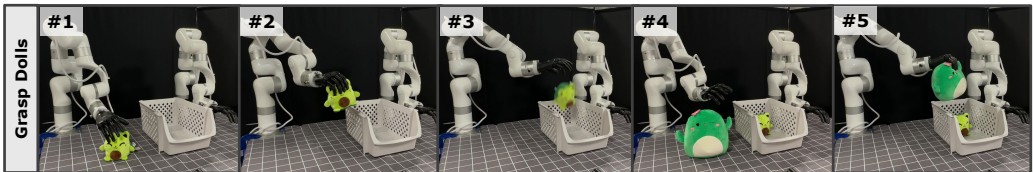

(d) The robot grasps the doll (*#1, #3* - Grasp) and places the doll into the basket (*#2, #4, #5* - Place).

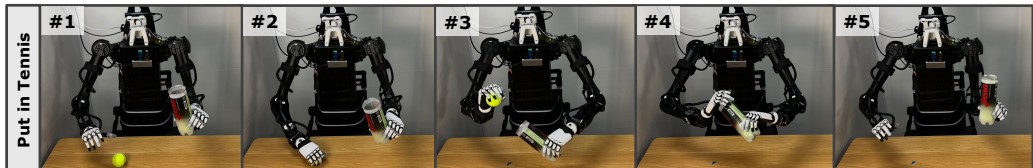

(e) The robot grasps the tennis ball (*#1-#3* - Grasp) and places the tennis ball back into the container (*#4, #5* - Place).

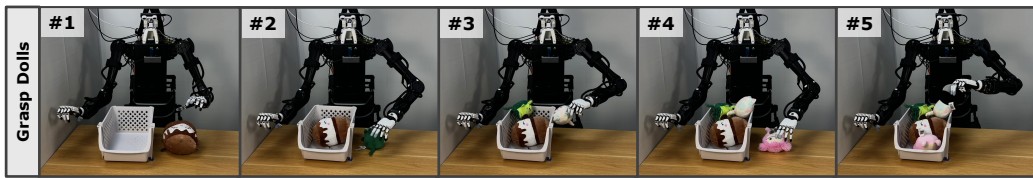

(f) H1 robot doing the same *Grasp Dolls* task as in Fig. 6d.

Figure 6: **Imitation Learning on Different Robots.** The first four tasks (*a-d*) involve imitation learning using the Xarm with the ability hand, while the last two tasks (*e, f*) are performed using the H1 with the inspire hand.

Table 3: **Imitation learning evaluation.** This table evaluates the system's ability to collect data and apply it to imitation learning tasks, followed by an evaluation of the performance in these tasks.

| Task | Robot | Hand | Data Collection | | Imitation | |
| --- | --- | --- | --- | --- | --- | --- |
| | | | Avg. Time (s) | Succ./Trials | Steps | Succ. Rate |
| Vacuum Keyboard | xArm7 | Ability | 45.6 | 30/38 | Grasp | 1 |
| | | | | | Activate | 0.6 |
| | | | | | Clean | 0.6 |
| | | | | | Place | 0.5 |
| Serve Candies | xArm7 | Ability | 37.6 | 45/53 | Grasp | 0.9 |
| | | | | | Open | 0.6 |
| | | | | | Throw Cap | 0.5 |
| | | | | | Pour | 0.4 |
| | | | | | Throw Tube | 0.4 |
| Wipe Whiteboard | xArm7 | Ability | 33.5 | 30/37 | Grasp | 1 |
| | | | | | Wipe | 0.7 |
| Grasp Dolls | xArm7 | Ability | 11.6 | 120/123 | Grasp | 0.9 |
| | | | | | Place | 0.8 |
| Put in Tennis | H1 | Inspire | 21.4 | 25/29 | Grasp | 0.9 |
| | | | | | Put in | 0.9 |
| Grasp Dolls | H1 | Inspire | 13.7 | 62/66 | Grasp | 0.9 |
| | | | | | Place | 0.9 |

## A   End-Effector Control Mapping Details

With our different modes of operation, we can accurately map the user's real-world end effector (EE) center, which can comfortably and freely draw spheres in physical space, to the center of the corresponding task's workspace, while adjusting the control scale.

---

**Algorithm 1** Mapping the Workspace

---

**Require:** Robot workspace center `robot_center`, robot workspace radius `robot_radius`
**Require:** Human movement data `max_x`, `max_y`, `max_z`, `min_x`, `min_y`, `min_z`
**Ensure:** Mapped human workspace within robot workspace
1: `human_center` $\leftarrow \left(\frac{max\_x + min\_x}{2}, \frac{max\_y + min\_y}{2}, \frac{max\_z + min\_z}{2}\right)$
2: `human_radius` $\leftarrow \min(max\_x - \text{human\_center.x}, max\_y - \text{human\_center.y}, max\_z - \text{human\_center.z})$
3: `control_scale` $\leftarrow \frac{robot\_radius}{human\_radius}$
4: `offset` $\leftarrow$ `robot_center` $-$ `control_scale` $\times$ `human_center`
5: `Mapped_ee` $\leftarrow$ `human_ee_position` $\times$ `control_scale` $+$ `offset`
6: Ensure $\|$`mapped_ee` $-$ `robot_center`$\| <$ `robot_radius` for safety

---

---

**Algorithm 2** Map Certain Task

---

**Require:** Robot workspace center `robot_center`, Robot workspace radius `robot_radius`
**Require:** Task center `task_center`, Task control scale `task_control_scale`
**Require:** Human movement data `max_x`, `max_y`, `max_z`, `min_x`, `min_y`, `min_z`
**Ensure:** Mapped human task within robot workspace
1: `human_center` $\leftarrow \left(\frac{max\_x + min\_x}{2}, \frac{max\_y + min\_y}{2}, \frac{max\_z + min\_z}{2}\right)$
2: `human_radius` $\leftarrow \min(max\_x - \text{human\_center.x}, max\_y - \text{human\_center.y}, max\_z - \text{human\_center.z})$
3: `control_scale` $\leftarrow$ `task_control_scale`
4: `offset` $\leftarrow$ `task_center` $-$ `control_scale` $\times$ `human_center`
5: `Mapped_ee` $\leftarrow$ `human_ee_position` $\times$ `control_scale` $+$ `offset`
6: Ensure $\|$`mapped_ee` $-$ `robot_center`$\| <$ `robot_radius` for safety

---

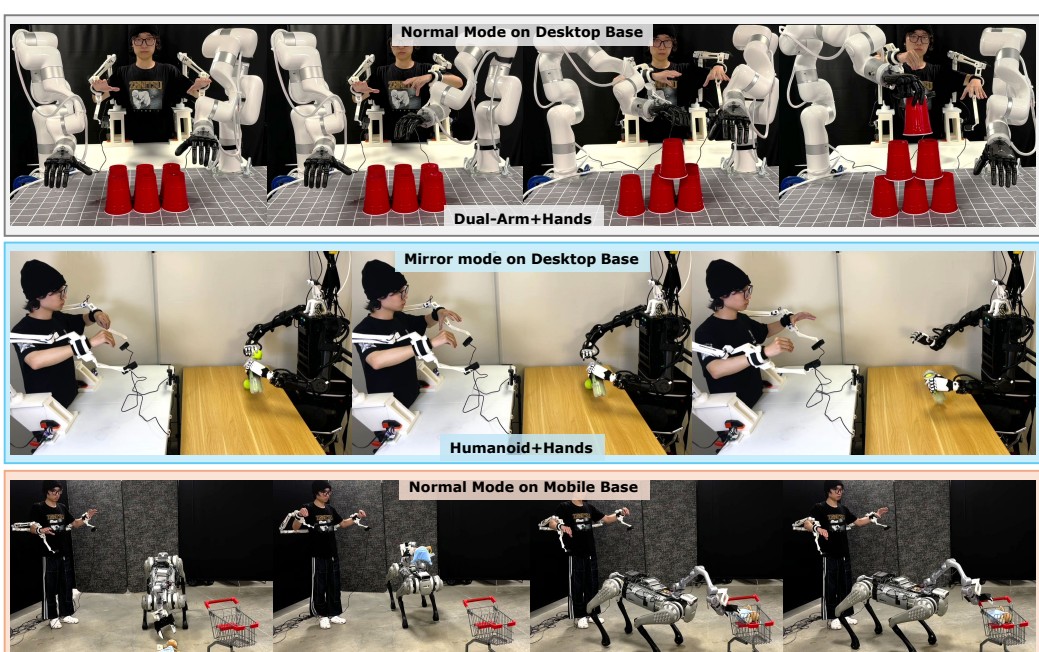

Figure 7: **Details of Cross-Platform Teleoperation.** This figure showcases various control modes for efficiently teleoperating different robots, including normal/mirror modes and hands/gripper configurations. Additionally, it illustrates the application of our Dual-base setup in various scenarios. In the *Dual-Arm+Hands* setup, normal mode with a desktop base is used to control the xArm with the ability hand. The *Humanoid+Hands* setup employs mirror mode with a desktop base to control the robot. In the *Quadrapeds+Gripper* setup, the right hand controls the robot in normal mode, while the left hand uses simple poses to manage the quadruped's movement using a pre-trained low-level control policy at a fixed velocity. The mobile base allows the operator to follow the robot's movements, enabling better control.

## B Hand Retargeting Details

We map the human hand pose data obtained from MediaPipe into joint positions of the teleoperated robot hand. This process is often formulated as an optimization problem [14], where the difference between the keypoint vectors of the human and robot hand is minimized. The optimization can be defined as follows:

---

**Algorithm 3** Map Rotation

---

**Require:** Task rotation range `task_roll_range`, `task_pitch_range`, `task_yaw_range`
**Require:** Human movement data `max_roll`, `max_pitch`, `max_yaw`, `min_roll`, `min_pitch`, `min_yaw`

1: $\texttt{scale\_roll} \leftarrow \frac{(\texttt{max\_roll}-\texttt{min\_roll})}{\texttt{task\_roll\_range}}$

2: $\texttt{scale\_pitch} \leftarrow \frac{(\texttt{max\_pitch}-\texttt{min\_pitch})}{\texttt{task\_pitch\_range}}$

3: $\texttt{scale\_yaw} \leftarrow \frac{(\texttt{max\_yaw}-\texttt{min\_yaw})}{\texttt{task\_yaw\_range}}$

4: $\texttt{offset\_roll} \leftarrow -\texttt{scale\_roll} \times \frac{(\texttt{max\_roll}-\texttt{min\_roll})}{2}$

5: $\texttt{offset\_pitch} \leftarrow -\texttt{scale\_pitch} \times \frac{(\texttt{max\_pitch}-\texttt{min\_pitch})}{2}$

6: $\texttt{offset\_yaw} \leftarrow -\texttt{scale\_yaw} \times \frac{(\texttt{max\_yaw}-\texttt{min\_yaw})}{2}$

7: $\texttt{Mapped\_roll} \leftarrow \texttt{human\_roll} \times \texttt{scale\_roll} + \texttt{offset\_roll}$

8: $\texttt{Mapped\_pitch} \leftarrow \texttt{human\_pitch} \times \texttt{scale\_pitch} + \texttt{offset\_pitch}$

9: $\texttt{Mapped\_yaw} \leftarrow \texttt{human\_yaw} \times \texttt{scale\_yaw} + \texttt{offset\_yaw}$

---

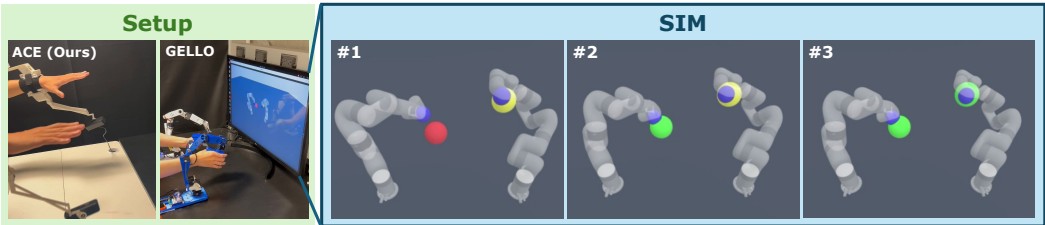

Figure 8: **Illustration of the Target-reaching Experiment.** We use xArm as an example. The blue spheres represent the end-effector, and the red and yellow spheres represent the targets for the left and right arms, respectively. When a target is successfully reached, it turns green to indicate success. We compare `ACE` (left) with GELLO (right). SIM #1 to #3 demonstrate the complete process of a task being completed.

$$\min_{q_t} \sum_{i=0}^{N} |\alpha v_{it} - f_i(q_t)|^2 + \beta |q_t - q_{t-1}|^2 \tag{2}$$

$$\text{subject to:} \quad q_l \leq q_t \leq q_u$$

where $q_t$ represents the joint positions of the robot hand at time step $t$, $v_{it}$ is the $i$-th keypoint vector for the human hand computed from the detected finger keypoints, $f_i(q_t)$ is the $i$-th forward kinematics function which takes the robot hand joint positions $q_t$ as input and computes the $i$-th keypoint vector for the robot hand, $q_l$ and $q_u$ are the lower and upper limits of the joint position, and $\alpha$ is a scaling factor to account for hand size difference. An additional penalty term with weight $\beta$ is included to improve temporal smoothness.

## C  Target-Reaching Task Details

We designed a target-reaching game to test the key performance parameters of both systems in simulation. A screenshot of the designed game is shown in Fig. 8, where the only goal is to control the end-effectors (also represented as spheres) to reach the randomly generated goals. When the goal for both arms is reached and the end-effectors are kept inside the goals for an assigned keep time, it is noted as a success and two new goals will be generated in the workspace. We take it as "inside" depending on the position of the end-effector and the target ball, using the following formula:

$$\mathbf{1}(\text{reached}) = \|p_{\text{ee}} - p_{\text{target}}\| <= r_{\text{target}} - r_{\text{ee}} \tag{3}$$

where $p$ represents for position and $r$ represents for radius, and $r_{\text{target}} > r_{\text{ee}}$ all the time. The game supports various types of robotics arms (with different kinetics), and during our test, we chose XArm for evaluation.

### C.1  Configurations

We construct four different game configurations by varying the workspace, the size of the end-effector/target balls, along the keep time. The four variants are designed to assess the teleoperation system's performance under different conditions by simulating real-world robotic tasks. These simulation settings correspond to real-world tasks ranging from highly precise operations like microsurgery to larger-scale tasks such as handling large objects. Specifically, these setups include:

The "Small & Small" setup mimics delicate tasks in confined spaces, while the "Large & Large" setup reflects operations requiring speed in a larger workspace. The "Medium & Small" and "Medium & Medium" settings represent tasks that balance precision and speed, such as electronics assembly or general manufacturing operations.

These variants differ in terms of workspace size, end-effector/target region size, and duration the end-effectors must remain in the goal regions. Their precise game parameters are shown in Tab. 4.

Table 4: The key parameters of the four variants. Note that the workspace, target, and end-effector are all represented as spheres so the only parameters are their radius.

| Workspace | Target | ACE Control Scale | Workspace Radius (cm) | Target Radius (cm) | End-Effector Sphere (cm) | Keep Time (secs) |
|---|---|---|---|---|---|---|
| Small | Small | 0.5 | 5 | 2 | 1 | 1 |
| Large | Large | 6 | 60 | 12 | 6 | 0.05 |
| Medium | Small | 1 | 10 | 4 | 2 | 0.5 |
| Medium | Medium | 2.5 | 25 | 8 | 4 | 0.1 |

## C.2 Participants

We invite 6 students as our volunteers, four of them are males and two of them are females. Their height ranges from 165 cm to 198 cm, and their weights vary from 55 cm to 90 kg.

## C.3 Experiment Details

**Playing process.** Each participant first plays general range 2 for 5 times, each time lasting 30 seconds. We have two objectives. First, to allow users to familiarize themselves with the controller through five relatively simple task scenarios. Second, to record the results of these five attempts to observe the learnability of the system Then, participants will play general 1, fine, and board for once. We test only once in other scenarios because both GELLO (the joint copy system) and ACE are highly intuitive teleoperation systems. We aim to see if users can quickly adapt when encountering a new scenario for the first time. Additionally, we want to observe the performance of the joint copy system and the ACE system in task scenarios with varying precision requirements.

**Grouping.** To ensure the fairness of the experiment and the comparability of the results, targets appear in the same positions within the identical workspace across tests. Familiarity with the testing simulation could also affect the outcomes. Therefore, we divided the six testers into two groups. One group completed the full ACE test before the *GELLO* test, while the other group did the opposite.

