# OpenReview forum: "ACE: A Cross-platform and visual-Exoskeletons System for Low-Cost Dexterous Teleoperation"
_robot-learning.org/CoRL/2024/Conference — CoRL 2024_

### Official Review · Reviewer_yoG4 · 2024-07-10
**Nice paper with very little learning**

**Originality:** 3
**Technical Quality:** 4
**Clarity Of Presentation:** 4
**Potential Impact:** 3
**Recommendation:** 2
**Confidence:** 3

**Review:**

The paper is well written and the presented hardware seems to be a valuable contribution to the robot learning community. The design decisions made by the authors are well explained and sound reasonable to me. The device is affordable (~600$) and seems to allow for accurate wrist pose tracking. Related work seems to be cited in in an adequate way.

Weaknesses of the paper in my opinion are:
- The paper is a hardware paper that uses learning only as a proof of concept. That means the authors showcase that their hardware can be utilized for collecting data that can be used for training imitation learning models. It really is a good paper but I don't think CoRL is the right conference for it. I'd prefer to see it at ICRA (or another conference that is more general or has a focus on robotics hardware) but it's up to the Area Chair to decide this.
- The experiments have a strong focus on the evaluation of the wrist pose tracking. The finger pose tracking has not really been evaluated.
- The authors mention that the precision of the wrist rotation tracking is limited. What does that mean? I wish the authors had discussed this in more detail.

**Quality Of The Limitations Section:**

2

**Questions For Rebuttal:**

- In l. 123 you mention that the two sources of information allow for tracking the hand pose relative to the world frame. Shouldn't that be the base frame of your device? While the transformation from the base of your device to the world frame is static for the desktop version, it's dynamic for the mobile version.

- Figure 3 (left) is too dark in my opinion. Would it be possible to increase the brightness?

- In l. 254+255 you say you have achieved 100% success rate for all tasks at the first step, which is not true according to Table 4. I suggest you either correct the sentence or modify it to make it more clear what you mean by "all tasks".

- Please elaborate more on why the precision of the wrist rotation tracking is limited and what could be done to alleviate this.

**Robotics Focus:**

4

**Summary Of Paper:**

The authors present a low-cost bi-manual teleoperation device which is comprised of an exoskeleton for tracking the operator's wrist poses and two cameras attached to the exoskeleton for tracking the operator's finger poses. The main purpose of the device is to provide affordable hardware for collecting data for robot learning, e.g., imitation learning approaches.

**Summary Of Recommendation:**

As I said, it is a good paper but I don't think CoRL is the right conference for it because the focus of the paper is not on learning but on actual hardware that might be of interest to the learning community.

---

### Official Review · Reviewer_14Vt · 2024-07-19
**Similar to existing solutions without sufficient novelty in algorithmic or implementation.**

**Originality:** 2
**Technical Quality:** 2
**Clarity Of Presentation:** 2
**Potential Impact:** 2
**Recommendation:** 1
**Confidence:** 4

**Review:**

This paper presents a low-cost exoskeleton design to capture bimanual object operation data from human demonstration. The authors claim that their system is a better design with specific considerations regarding pose estimation and end-effector mapping. The authors did some preliminary experiments to verify its basic functionality in teleoperation various robot system. They also compared in simulation against GELLO with superior performance. They did not discuss in detail regarding the limitation of this work.
1.	The paper is written in general descriptive language without too much technical details, making this paper easy to read. However, considering this is already a very hot topic with various research interests, it is crucial for the authors to avoid using similar descriptive language and use more technical analysis and derivation to justify why the proposed system is better in what way and why.
2.	The authors raised a very interesting question at the beginning of this paper but did not provide any follow-up research or analysis into the particulars. And based on the current manuscript, I did not see any clear answer or in-depth analysis.
3.	The strength of this work is probably the various hardware platform test they performed, which could be very challenging for most labs. The device takes an exoskeleton form, which is wearable by design. And the data collected is anthropomorphic instead of 2 finger gripper.
4.	The weakness of this work is mainly on the lack of technical details, lack of in-depth analysis regarding why it performs better, lack of algorithmic related research or insights in the current format.

**Quality Of The Limitations Section:**

2

**Questions For Rebuttal:**

1.	So, what is the answer to the question the authors raised in the beginning? “What key information must be captured to effectively perform  bimanual dexterous manipulation tasks?” Are the authors’ finding the same or different from other authors? It seems this paper gradually drifts from one theme to another while reading it.
2.	The majority of this paper lacks sufficient technical details for a thorough evaluation and most of the content are descriptive of the general process without further analysis or technical details. Please consider focus on at least one thing to explain or describe clearly what’s the advantage of the proposed work, why it works, how better it performs for robot learning task.
3.	Comparison with GELLO in simulation lacks sufficient technical details or in-depth analysis to justify why the result obtained. It is understandable if the authors do not have a hardware GELLO, so choosing to investigate through simulation. But this section lacks sufficient technical details on settings of the simulation parameters, which may vary significantly with a strong bias. And the authors did not provide any in-depth explanation why one performs better than the other, limiting the potential impact of this work.
4.	Table 2 is highly biased, and I wonder if the system requires a certain range of motion at the waist to be functional, even it can be used as a wearable device.

**Robotics Focus:**

4

**Summary Of Paper:**

This paper presents a low-cost exoskeleton design to capture bimanual object operation data from human demonstration.

**Summary Of Recommendation:**

No clear novelty but more of an increamental work withouth sufficient algorithmic or system contribution comparing to existing work.

---

### Official Review · Reviewer_BGSU · 2024-07-19

**Originality:** 3
**Technical Quality:** 2
**Clarity Of Presentation:** 3
**Potential Impact:** 3
**Recommendation:** 3
**Confidence:** 4

**Review:**

Strengths:
1. It is an interesting exoskeleton system for teleoperation, which can be applied to multiple types of bimanual dexterous robots.
2. The system is low-cost and can be easily manufactured and distributed more widely.
3. Good experiment results on both humanoid robots and robot arms.


Limitations:
1. Although it is a cool design for switching between desktop and mobile versions, I’m curious if we really need it. Since the system is designed for “tele”operation, which is supposed to be used remotely, it shouldn’t really matter whether the system is fixed on a desktop or mobile, right?
2. The system relies on two hand-facing cameras to get the dexterous finger motion, which are mounted quite far from the human hand. Do the cameras collide with each other during teleoperation? For example, in Figure 1 Mobile version H1, it looks like the cameras has collision.
3. One important missing related work is [1]. Both systems are exoskeletons, but [1] focuses on collecting manipulation data directly without a robot, while this work focuses on teleoperation. I think both problems are important, but teleoperation seems to be a subset of [1], and [1] covers more use cases (teleoperation and collecting data without a robot). I hope to see the authors provide more discussion on this related work to justify the novelty of the proposed system.
4. It’s cool to see real-world bimanual manipulation results, but most of the tasks are pick-and-place style. What kind of new capability can this system provide compared to other teleoperation methods like VR headsets [2]? Also, it seems like the exoskeleton does not match with the robot’s joint design, which means it cannot perform joint mapping like ALOHA or GELLO. Does that affect teleoperation capability in more challenging tasks?
5. L249: “It is important to note that because we can collect a large amount of data in a short time, our GraspAndPlaceDolls policy has generalizability.” Are you trying to say that because data collection is fast, you can collect more data within the same amount of time, so your model performs better? But why does it also generalize better?
6. It would be nice to have a systematic measurement of the accuracy of the system. Figure 4 is good, but it contains compounding errors combined with the robot control error. Could you provide a number of distances with mean and standard deviation about the teleoperation system only? It would be helpful for readers to understand the quality of the data.
7. Is there any specific reason to use point clouds as input instead of images? It seems like an orthogonal problem to the main focus of this work (teleoperation system), which makes me curious about the reason for this experiment setup.

[1] AirExo: Low-Cost Exoskeletons for Learning Whole-Arm Manipulation in the Wild

[2] Open-TeleVision: Teleoperation with Immersive Active Visual Feedback

**Quality Of The Limitations Section:**

2

**Questions For Rebuttal:**

same as the limitation section.

**Robotics Focus:**

4

**Summary Of Paper:**

The paper introduces ACE, a low-cost and versatile teleoperation system for bimanual dexterous manipulation. ACE employs two bimanual exoskeleton arms and hand-facing cameras to capture 3D hand poses and end-effector positions in real-time. The system's modular design allows for cross-platform operations, demonstrated across six different robot platforms. ACE can efficiently gather teleoperation data and train imitation learning policies for manipulation tasks.

**Summary Of Recommendation:**

Overall, this is an interesting system for teleoperation. However, more discussion of prior exoskeleton work is needed. Additionally, the authors need to clarify the new capabilities this system provides compared to prior teleoperation methods.

---

### Author Rebuttal · Authors · 2024-08-13

### Rebuttal Zip File

- **Revised Paper**: Modified parts are highlighted in red.
- **Revised Appendix**: Modified parts are highlighted in red.
- **Rebuttal Video**: Includes additional experiments demonstrating FK accuracy, teleoperation, and learning results.

Please also see the comments to individual reviewers for more details.

---

### Decision · Program_Chairs · 2024-09-04

**Decision:**

Accept

**Comment:**

The authors propose a low-cost system for teleoperation. The reviews largely agree the system is interesting, but there are many missing details with respect to implementation and efficacy.

The authors propose a low-cost teleoperation framework which can be used to collect data for many different types of robots, including humanoids. Hardware like this is essential to robot learning, and they attempt to make the system reproducible and open-source. This type of systems work is valuable to the community and I think it's worth endorsing, even if it does not immediately show exciting new learning algorithms in action. The quality of the writing is good.

The authors provide some details about how to reproduce and build their system. They provide detailed feedback on how their method is an improvement over the state of the art.  They also provided a large number of key details during the rebuttal which remove some concerns about clarity and reproducibility, and they show interesting tasks like spraying and pipetting which would be very difficult for well-known humanoid teleop systems based on video capture.